# How Employee’s Leadership Potential Leads to Leadership Ostracism Behavior: The Mediating Role of Envy, and the Moderating Role of Political Skills

**DOI:** 10.3390/ijerph17093080

**Published:** 2020-04-28

**Authors:** Ying Xue, Xiyuan Li, Hongmei Wang, Qiu Zhang

**Affiliations:** Economics and Management School, Wuhan University, Wuhan 430072, China

**Keywords:** leadership potential, envy, leadership ostracism behavior, political skills

## Abstract

Recently, research on the leadership potential of employees has gradually attracted the attention of scholars. However, further exploration is required to better understand the upward influence of employee’s leadership potential on their leaders. This study examined the mechanisms behind the impact of employee’s leadership potential on leadership ostracism behavior. Moreover, the mediating role of leader’s envy and the moderating role of employee’s political skills in the relationship between employee’s leadership potential and leadership ostracism behavior were investigated. The results of an empirical analysis of 221 employee–leader pairs, studied over multiple periods, are as follows: employee’s leadership potential had a significant positive impact on leader’s envy and leadership ostracism behavior; leader’s envy had a significant positive impact on leadership ostracism behavior; and leader’s envy mediated the relationship between leadership potential and leadership ostracism behavior. In addition, employee’s political skills negatively moderated the indirect effect of leadership potential on leadership ostracism behavior through leader’s envy. The leadership potential of employees with more political skills appeared to have less influence on organizational ostracism via leader’s envy. This study explored the “dark-side” of employee’s leadership potential by understanding its impact on their leaders; the findings have theoretical and practical significance for companies.

## 1. Introduction

With the deepening of globalization and artificial intelligence, human capital has become a core competency for enterprises, managers have paid increased attention to the interaction effects among employees [1]. In order to attain sustainable development, enterprises have been devoted to the identification, training, and development of corporate leaders and have paid more attention to occupational and mental health of employees. Bersin and Chamorro-Premuzic [2] pointed out that, in the past, companies tended to promote employees based on outstanding achievements and performance; however, these employees were usually found to lack leadership qualifications after they were promoted. For that reason, it was suggested that if a company intends to maintain a strong development trajectory, managers should not only consider the performance of their employees when giving promotions, but also their potential as leaders [2]. Employees with leadership potential grow faster and possess traits that match the development of the company. They; therefore, exist as the core human resource in the HRM system and the key to achieving long-term sustainable development. Hence, enterprises should identify and coach employees with leadership potential so as to ensure a continuous provision of competitive leaders to further promote their sustainable development and the enterprises are suggested to inhibit harmful workplace behaviors in order to improve the occupational health.

Currently, there are limited studies on employee’s leadership potential, and the majority of existing studies focus on identifying leadership potential [3] and the impact of such potential on the employees themselves and their colleagues [4]. Few studies have yet to explore the impact of employee’s leadership potential on their leaders. Employees with leadership potential are important resources within an organization and are important to its sustainable development; therefore, they are likely to have a significant impact on managers within an organization. On that account, studying the impact of employee’s leadership potential on their leaders would reveal the negative impact of employees’ leadership potential on leaders and enrich the current research in the corresponding academic fields. Additionally, in the researches on the antecedents of leadership ostracism behavior, most of literatures suggest that low-performing employees are more likely to suffer ostracism [5]. Only a few studies point out that employees with outstanding performance may also be the target of ostracism. Therefore, this study can further verify the prior researches.

Therefore the purposes of this study are as follows: (1) To investigate the impact of employee’s leadership potential on leadership ostracism behavior; (2) to examine the mediating role of leader’s envy in the relationship between employee’s leadership potential and leadership ostracism behavior; and (3) to analyze the influence of employee’s political skills on the above relationships in order to find out the boundary condition of the impact that employee’s leadership potential has on leadership ostracism behavior. This study begins with theoretical discussions and hypotheses development. The following part introduces the methodology of this study. Thirdly, the empirical results are presented. Following that, the theoretical implications, practical implications, and limitations are discussed and future directions are recommended. Finally, this study will end with a brief conclusion.

Based on the empirical investigation of 221 employee–leader pairs, this study constructed a theoretical model of the relationships between employee’s leadership potential, leader’s envy, leadership ostracism behavior, and employee’s political skills. Then, emotional pathway analysis was adopted to determine the mechanisms behind the influence of employee’s leadership potential on leadership ostracism behavior. The findings uncovered how employee’s leadership potential could have a negative impact on the organization.

## 2. Theory and Hypotheses

### 2.1. Theoretical Discussion

Social comparison theory refers to people continually make comparison with other people to assess their own opinions and abilities [6]. The theory includes both upward and downward social comparisons [7,8]. Social comparisons to someone who is performing better can be regarded as an upward comparison, contrarily, social comparisons to someone who is performing worse can be regarded as a downward comparison. Early researches have shown that both upward comparison and downward comparison can result in either positive outcomes [9,10,11] or impaired outcomes [12,13]. This theory also points out that self-evaluation is a powerful intrinsic motivator; as such, social comparisons are a common phenomenon [6]. During the social comparison process, a range of emotional experiences emerge, which in turn trigger corresponding behaviors [14,15]. 

Protection motivation theory points out that external stimuli (such as pressure or threat) can prompt individuals to turn on the self-protection mechanism and then lead to the corresponding self-protection behavior [16]. For instance, a research found out that the pressure of leaders in workplace will lead to the generation of negative emotions, and the negative emotions will lead to the self-protection mechanism of leaders and the implementation of harmful behaviors [17]. Previous researches have supported that the ability of employees can be a major trigger for leaders’ negative emotions [18,19]. According to protection motivation theory, when leaders experience negative emotions or feelings, they will generate corresponding behaviors to protect themselves [17].

### 2.2. Employee’s Leadership Potential and Leadership Ostracism Behavior

Employee’s leadership potential refers to the potential of junior employees to be developed into leaders [20,21]. Based on the analysis of 40 influential journal papers, scholars summarized the four dimensions of leadership potential as “analytical skills, learning agility, drive, and emergent leadership” [3]. Current research on employee’s leadership potential tends to focus on identifying leadership potential [3] and how such potential impacts the employees themselves and their colleagues [4]. However, employees with leadership potential are an important resource within an organization and a critical promoter of sustainable growth; hence, they also have a significant impact on organization management.

Ostracism refers to employees’ perceived interpersonal deviance from their leaders and can be seen as intentional or unintentional ostracism, open or discreet neglect, rejection, and exclusion in the workplace [22,23]. Different sources of exclusion have different impacts on employee attitudes and work behaviors [24]. Compared to other sources of exclusion, due to the significance of management positions in an organization and ownership of organizational resources, exclusion from managers was found to have a more prominent negative impact on the physical and mental health and work behaviors of employees. Based on previous literatures, the factors which can affect leadership ostracism behavior can be summed up in three main factors, which include factors of leaders [25,26], factors of ostracized employees [18,19], and factors of organizations [27]. Because employees with high leadership potential have a significant impact on organizations and leaders, this study focuses on exploring one of the factors of ostracized employees—employee’s leadership potential’s effect on leadership ostracism behavior.

The majority of existing studies suggest that low-performing and undervalued employees are more likely to suffer ostracism [5]. However, employees with leadership potential may also be the target of ostracism or even suppression from managers due to their outstanding performance. It is possibly because the employees or the informal leaders with high leadership potential can be strong competitors for leaders’ status and power [18]. This kind of unfavorable social comparison could make leaders perceive the lack of power and regard the employees with high leadership potential as a threat to their authority and position [19]. According to the conservation of resources theory [28], individuals have a tendency to preserve, protect, and obtain resources; moreover, when they expect resources to be threatened, individuals are likely to use existing resource reserves to adopt active adaptive strategies to prevent the further loss of resources. As conditional resources, managers’ status in an organization and corresponding authority give them the ability to control and influence others [29]. When a manager perceives that his or her status or power is threatened, he or she may manifest coercive and negative behaviors with the expectation to restore status and power [30]. Employee’s leadership potential is likely to be perceived as a threat to a manager’s status and power and may thus trigger corresponding action. The leaders may neglect and alienate the employees, impede employee’s promotion, reduce training opportunities, and perform other ostracism behavior to hinder this employee’s further success. Hence, based on the above analysis, hypothesis 1 was proposed:

**Hypothesis** **1** **(H1).**
*Employee’s leadership potential positively affects leadership ostracism behavior.*


### 2.3. The Mediating Role of Leader’s Envy

Managers enjoy privileges and advantages alongside their status in an organization. The benefits obtained by managers include valuable extrinsic rewards, autonomy in decision-making, control of resources, and opportunities to connect with external power holders [31,32]. These benefits are drivers that motivate many individuals to pursue managerial positions [33]. Despite the inherent advantages of managerial roles, evidence suggests that managers also envy subordinates [18,34,35]. Envy is most likely to occur when subordinates have strong social skills, show leadership potential, have close relationships with senior management, and are considered to be the main source of corporate innovation and progress [36].

According to social comparison theory, self-evaluation is a powerful intrinsic motivator; as such, social comparisons are a common phenomenon [6]. During the social comparison process, a range of emotional experiences emerge, which in turn trigger corresponding behaviors [14,15]. Specifically, positive emotions trigger pro-social behaviors, while negative emotions trigger antisocial behaviors in an organization [37]. Studies have found that unfavorable upward social comparison is more likely to cause feelings of envy. Specifically, while forming comparisons, it is more common to develop feelings of envy toward individuals that are similar to oneself but have certain advantages in key areas [38,39,40,41]. As a double-faceted emotional variable, envy not only highlights what an individual lacks in him or herself (own disadvantages), but also what the compared party has that oneself does not (advantages of the compared party). Focusing on one’s own disadvantages may lead to self-abasement and negative emotions, whereas focusing on the advantages of others may lead to the evaluation that the compared party is not worthy of having such advantages, thereby leading to feelings of aversion and anger [42].

Due to the prevalence of social comparisons, and in order to evaluate power and status, it is logical that managers are also making social comparisons with their subordinates, especially subordinates that are seen as superior to others, such as those with high leadership potential. Furthermore, empirical data have shown that adverse downward comparisons are common in the workplace and trigger feelings of envy, and leaders tend to make downward comparison with employees that possess specific superior qualities [18,35,36]. Subordinates with superior qualities can be defined as having strong social skills, the potential to become leaders, good relationships with senior managers, or as sources of innovative ideas [36]. The adverse social comparison between leaders and employees could make leaders perceive the lack of power and regard the employees with high leadership potential as a threat to their authority and position and leads to the feelings of envy and insecurity [19]. For that reason, it can be assumed that, during the process of social comparison between managers and employees, employees’ display of leadership potential may trigger downward feelings of envy in their leaders. Such downward envy is reflected as envy toward both the specific quality and the employee. Hence, the following hypothesis was proposed:

**Hypothesis** **2** **(H2).**
*Employee’s leadership potential has a positive effect on leader’s envy.*


According to the definition proposed by Parrott and Smith [43], envy occurs when an individual perceives that he or she lacks a superior quality, achievement, or possession that is possessed by others. Envy stemming from social comparison is a distressing experience that threatens a person’s self-concept. Research has found that envy can lead to harmful behavior toward the object of one’s envy. In addition, someone may attempt to suppress the other party through certain behaviors and to reduce the perceived dissonance between the compared party and oneself [14,44,45,46,47,48]. Based on protection motivation theory [16,17], fear and negative emotions are generated when an individual feels threatened, which may initiate a self-protection mechanism that leads to corresponding protective behaviors. Therefore, the present authors proposed that leader’s envy could be used as a positive predictor for ostracism. Specifically, when a manager envies his or her subordinate, he or she is more likely to adopt harmful behaviors, such as ostracism, as a self-defense mechanism. Hence, based on the discussion above, hypothesis 3 was proposed:

**Hypothesis** **3** **(H3).**
*Leader’s envy has a positive effect on leadership ostracism behavior.*


According to the previous discussions, it can be displayed that the leaders may feel envy when they make an unfavorable social comparison with the high leadership potential employees, and leads to leadership ostracism behavior further. Hypothesis 2 discussed the positive relationship between employee’s leadership potential and leader’s envy. Hypothesis 3 discussed the positive relationship between leader’s envy and leadership ostracism behavior. Therefore, based on Hypothesis 2 and Hypothesis 3, we further suggested that employee’s leadership potential stimulates the envy of their leaders, leading leaders to ostracize their employees as a means of self-protection; hence, employee’s leadership potential would cause leader’s envy, which thereby leads to leadership ostracism behavior. Thus, the following hypothesis was proposed:

**Hypothesis** **4** **(H4).**
*Leader’s envy mediates the relationship between employee’s leadership potential and leadership ostracism behavior.*


### 2.4. The Moderating Role of Employee’s Political Skills

Nevertheless, not all employees with high leadership potential trigger envy from their leaders. Situational factors, such as organizational culture, and individual factors, such as employee and manager personality traits, also affect the relationship between employees’ leadership potential and leader’s envy. From the perspective of organizational politics, the present authors see political skills as individual characteristics that significantly affect the relationship between employees’ leadership potential and leader’s envy.

Political skills refer to the ability to effectively understand others in the workplace and to utilize such knowledge to create influence that strengthens individual or organizational goals [49]. According to Ferris, Treadway, Kolodinsky, Hochwarter, Kacmar, Douglas and Frink [49], political skills include the following four dimensions: apparent sincerity, social astuteness, interpersonal influence, and networking ability. The four dimensions have their own distinctive constructs yet are correlated to one another. Individuals that master the skill of apparent sincerity give the appearance of being honest, kind, and trustworthy, which affects the intentions or motives for their behaviors as perceived by others. Social astuteness refers to the ability to astutely observe other individuals and oneself, as well as to accurately interpret and analyze the behavior of others. Interpersonal influence tends to give the impression of having a humble, pleasant, and convincing personal style, which exerts a strong influence over others and oneself. Networking ability refers to the skill of identifying, developing, and utilizing various relationship networks. 

Scholars point out that individuals with more political skills are better able to accurately interpret social situations and adjust their behavior, accordingly, choosing appropriate methods and strategies to influence others [49,50,51]. According to Mintzberg [52], individuals with more political skills are better at influencing others through persuasion, manipulation, and negotiation. Therefore, employees with more political skills are more likely to understand and perceive the potential negative impact of their leadership potential on their leaders, such as potential negative emotions and behaviors. Hence, it is reasonable to believe that they would be better at using such knowledge to avoid potential risk and problems by taking different measures to achieve personal or organizational goals. Employees with high leadership potential and more political skills should; therefore, be able to perceive the impact of such potential and take corresponding measures to avoid negative outcomes (such as leader’s envy), by demonstrating their value without triggering a loss of pride from managers, or by using persuasion and consultation to communicate their viewpoints to ensure support from their leaders. Thus, when employees exert political skills, their high leadership potential should be less likely to lead to envy, thereby reducing the likelihood of ostracism. Thus, hypothesis 5 was proposed:

**Hypothesis** **5** **(H5).**
*Employee’s political skills negatively moderate the impact of employee’s leadership potential on leadership ostracism behavior via leader’s envy. Specifically, employees with more political skills are less likely to lead to the indirect impact that their leadership potential has on ostracism via leader’s envy.*


According to the discussion above, the following conceptual model has been conducted (See Figure 1).

## 3. Materials and Methods

### 3.1. Sample and Data Collection

The data of this study originated from 12 enterprises in Shenzhen, Shanghai, Wuhan, and Beijing, covering industries such as education, finance, technology, design, and construction. In order to avoid common method bias, this study adopted a multi-source and multi-period data collection method where leaders and employees were asked to complete the questionnaire separately. Three surveys were conducted, with intervals of one month between each survey. In the first period (T1), leaders and employees were asked to provide information related to their demographic characteristics, such as gender, age, and educational background. In addition, leaders were asked to evaluate their subordinates’ leadership potential, and employees were asked to report their political skills. In the second period (T2), leaders were asked about their feelings of envy toward their subordinates. In the third period (T3), employees were asked to evaluate any ostracism they faced from their leaders.

In T1, the questionnaires were distributed to 300 employee–leader pairs, and 276 pairs of valid responses were recovered. In T2, the questionnaires were distributed to the above 276 employee–leader pairs, and 248 pairs of valid responses were recovered. In T3, the questionnaires were distributed to the above 248 employee–leader pairs, and 221 pairs of valid responses were recovered (recovery rate = 73.7%). Hence, the 221 valid response pairs were used for data analysis and hypothesis testing. Among the employee participants, 57.9% were male and 42.2% were female, and 90.1% received a bachelor’s degree or higher. Among the leader participants, 62.5% were male and 37.5% were female, and 97.5% received a bachelor’s degree or higher (See Table 1).

### 3.2. Measures

The scales used to measure the variables in this study were as follows:

(1) Employee’s Leadership Potential (T1)

The 4-item leadership potential scale developed by Mueller, Goncalo and Kamdar [53] was utilized to measure employee’s leadership potential. Leader participants were asked to rate the leadership potential of their subordinates on a 7-point Likert scale (“1” = “totally disagree” and “7” = “totally agree”). Item examples include “I think he/she has the potential to be an effective leader” and “I think he/she has the potential to advance to a leadership position.”

(2) Leader’s envy (T2)

The 4-item scale developed by Kim and Glomb [47] was introduced to measure leader’s envy. Leader participants were asked to truthfully report their envy toward their subordinates on a 7-point Likert scale (“1” = “totally disagree” and “7” = “totally agree”). An item example includes “It is so frustrating to see this person succeed so easily!”

(3) Employee’s Political Skills (T1)

The 6-item Political Skill Inventory developed by Ferris, Treadway, Kolodinsky, Hochwarter, Kacmar, Douglas and Frink [49] was adopted to measure employee’s political skills. Employee participants were asked to self-assess their political skills on a 7-point Likert scale (“1” = “totally disagree” and “7” = “totally agree”). Item examples include “I spend a lot of time and effort at work networking with others” and “I am good at using my connections and networks to make things happen at work.”

(4) Leadership Ostracism Behavior (T3)

The leadership ostracism behavior scale, as adopted from O’reilly, Robinson, Berdahl and Banki’s study [54], which had six items. Employee participants were asked to evaluate the frequency of their leaders’ ostracism behavior on a 7-point Likert scale (“1” = “never” and “7” = “always”). Item examples include “your leader ignored or failed to respond to your message” and “your leader excluded you from influential roles or committee assignments.”

(5) Control Variables (T1)

Participants’ demographic variables (gender, age, and educational background) were introduced as the control variables. Specifically, the participants were divided into four age groups (equal to or younger than 30 years old, 31 to 40 years old, 41 to 50 years old, and equal to or older than 50 years old); a dummy variable was used to represent gender (“1” = “male” and “0” = “female”); and the participants’ educational backgrounds were divided into senior high school or lower, bachelor’s degree, and master’s degree and higher. The reliability and validity of the scales are exhibited in Table 2.

### 3.3. Reliability and Validity Analysis

Internal consistency reliability (Cronbach’s α) and composite reliability (CR) were used to evaluate the reliability of the employee questionnaire used in this study. Fornell and Larcker [55] suggested that CR should be above 0.6 for an instrument to be considered to reliable. It can be seen from Table 2 that the internal consistency reliability and CR of both scales were greater than 0.8, indicating that the employee questionnaire had satisfactory reliability.

Next, confirmatory factor analysis (CFA) was employed to test the discriminant validity of the scales. As shown in Table 3, the goodness of fit of employees’ leadership potential, leader’s envy, leadership ostracism behavior, and Employee’s Political Skills was better than the other models (χ2 = 839.764, df = 458, RMSEA = 0.062, CFI = 0.937, and TLI = 0.932), while the goodness of fit of other models was not satisfactory. These findings showed that the scales of the four variables had satisfactory discriminant validity. In addition, Fornell and Larcker [55] suggested that an average variance extracted (AVE) value above 0.5 indicates that the scale has satisfactory convergence validity. Table 2 shows that the values of the AVE of the variables were all greater than 0.500, showing that the scales had good convergence validity.

## 4. Results

### 4.1. Descriptive Statistics

Table 4 exhibits the mean, standard deviation, and correlation coefficient of each of the variables. As can be seen from Table 4, employees’ leadership potential had a significant positive correlation with leader’s envy (*r* = 0.428; *p* < 0.01) and organizational ostracism (*r* = 0.288; *p* < 0.01). In addition, leader’s envy and ostracism were also positively correlated (*r* = 0.380; *p* <0.01). These results initially supported the hypotheses proposed by the study.

### 4.2. Hypotheses Testing

SPSS was used for the linear regression analysis to explore the influence of employees’ leadership potential on leaders’ ostracism behaviors (H1). The results presented in Table 5 showed that *F* = 19.878 (*p* < 0.001), indicating that the independent variable had a significant effect on the dependent variable. In addition, *R*^2^ = 0.083, showing that the independent variable could explain 8.3% of the variances in the dependent variable. Furthermore, *β* = 0.228 (*p* < 0.01), confirming that employees’ leadership potential had a significant positive effect on leadership ostracism behavior. Therefore, H1 was supported.

The testing procedures of H1 were used to explore the influence of employees’ leadership potential on leader’s envy (H2). The results presented in Table 6 showed that *F* = 48.984 (*p* < 0.001), indicating that the independent variable had a significant effect on the dependent variable. In addition, *R*^2^ = 0.183, revealing that the independent variable could explain 18.3% of the variances in the dependent variable. Moreover, *β* = 0.428 (*p* < 0.001), confirming that employees’ leadership potential had a significant positive effect on leader’s envy. Hence, H2 was supported.

Next, the same approach was adopted to examine the relationship between leader’s envy and leadership ostracism behavior (H3). The results presented in Table 7 showed that *F* = 37.051 (*p* < 0.001), indicating that the independent variable had a significant effect on the dependent variable. In addition, *R*^2^ = 0.145, showing that the independent variable could explain 14.5% of the variances in the dependent variable. Furthermore, *β* = 0.380 (*p* < 0.001), confirming that leader’s envy had a significant positive effect on leadership ostracism behavior. Hence, H3 was supported.

Next, the bootstrapping method was introduced to investigate the mediating effect of leader’s envy (H4). Process, a plug-in for SPSS developed by Hayes [56], was employed to conduct repeated sampling 5000 times. It can be seen from Table 8 that the 95% confidence interval for bias correction of the indirect effect of employees’ leadership potential on leadership ostracism behavior was between [0.057, 0.195] (which did not include 0); and that the direct effect between the two variables was between [0.017, 0.258] (which did not include 0). These findings indicated that both the direct and indirect effects of the independent variable on the dependent variable were significant; leader’s envy played a mediating role between employees’ leadership potential and leadership ostracism behavior. Hence, H4 was supported.

In the next step, employee’s political skills were introduced as a moderating variable of the first half of the mediating effect (H5). As shown in Table 9, the interaction term of employees’ leadership potential and political skills was statistically significant (*t* = −2.535, *p* < 0.05). A simple slope diagram was drawn based on the results of the regression analysis (see Figure 2), which demonstrates that, compared to employees with few political skills, the leadership potential of employees with more political skills tended to have significantly less impact on leader’s envy. These results indicated that the simple moderating effect of employee’s political skills on the relationship between employees’ leadership potential and leader’s envy was prominent.

Table 10 showed that, under the bootstrap 95% confidence interval, the moderating effect of political skills of participants with few political skills (M-1SD) and moderate political skills (M) on the mediating effect of leader’s envy was not substantial; the 95% confidence intervals were [−0.056, 0.125] and [−0.102, 0.078], respectively. However, the moderating effect of the political skills of participants with more political skills (M+1SD) on the impact that their leadership potential had on leadership ostracism behavior via leader’s envy was significant; the 95% confidence interval was between [−0.172, −0.051] (which did not include 0). In summary, employee’s political skills negatively moderated the indirect impact that their leadership potential had on leadership ostracism behavior via leader’s envy. When employees possessed more political skills, their leader’s envy tended to be lower, leading to reduced leadership ostracism behavior. Therefore, H5 was supported.

## 5. Discussion

### 5.1. Theoretical Implications

This study examined the impact of employee’s leadership potential on leadership ostracism behavior and the mechanisms behind such an impact. The findings had the following theoretical implications.

Firstly, the study expanded the scope of research on employee’s leadership potential and established connections between such potential and leadership ostracism behavior. There are limited studies on employee’s leadership potential, and the majority of the existing studies have tended to discuss the impact on employees themselves and their colleagues; few studies have explored the upward impact on their leaders [4]. The results showed that employee’s leadership potential could lead to negative behaviors from managers, such as leadership ostracism behavior, and revealed how leadership potential can have a negative effect on organizational performance. In addition, the findings of this study enriched the research on the negative behavior of leaders. The majority of past studies have tended to focus on the likely outcome variables of leaders’ negative behavior as well as the corresponding mechanisms, and few studies have focused on the causal variables of such behavior. This study has found that personal abilities may also be a trigger of leaders’ negative behavior. The findings enriched current research on the causal variables of leaders’ negative behaviors. 

Secondly, this study enriched the research on the mediating mechanism of employee’s leadership potential. There is currently limited researches on the mediated variables of leadership potential. This study revealed the mediating mechanism between employees’ leadership potential and leadership ostracism behavior through emotional paths. Based on social comparison theory and protection motivation theory, the study explored the mediating role of leader’s envy in the relationship between employees’ leadership potential and leadership ostracism behavior. Additionally, the mediating effect of envy validated past studies on leader’s envy, and promoting further research into the role of emotions in the workplace. 

Lastly, this study uncovered the moderating role of employee’s political skills. The findings showed that employee’s political skills effectively alleviate the relationship between employee’s leadership potential and leader’s envy and moderates the effect that leadership potential has on leadership ostracism behavior through leader’s envy. The empirical results revealed the positive role of employee’s political skills and explored the boundary conditions of the effect of employee’s leadership potential.

### 5.2. Practical Implications

The empirical results had the following meaningful implications in practice. Firstly, our findings revealed that employee’s leadership potential may lead to leader’s ostracism behavior. Hence, organizations should carefully consider personal characteristics and emotional traits when selecting and appointing managers; implement clear regulations and penalty policies, provide training, conduct workshops, and socialization so as to mitigate the negative outcomes of the leadership ostracism behavior; adjust leader’s reward and incentive mechanism, part of leader’s compensation can be taken depends on the improvement of employee’s leadership potential, thereby leaders would internally recognize employee’s leadership potential and reduce ostracism behavior.

Secondly, this study explored the mediating role of leader’s envy. Hence, organizations are suggested to promote the consistency of work objectives by job design. In this way, the communications between leaders and employees can be promoted and the harmful emotion can be reduced sequentially. Additionally, leaders need to understand envy can be harmful for their occupational health and mental health. Hence, they are suggested to do some personality development trainings to reduce their envy and develop the ability of empathy.

Lastly, employee’s political skills can weaken the relationship impact of employee’s leadership potential on leadership ostracism behavior via leader’s envy. Therefore, organizations are suggested to focus on recruiting employees with more political skills and implement corresponding training to improve the political skills of corresponding candidates. Additionally, employees should promote the political skills and improve interpersonal relationships to minimize the harmful effects on occupational health and mental health.

### 5.3. Limitations and Future Research Directions

Although this study had the above theoretical and practical implications, certain limitations existed. Initially, although this study adopted a multi-source and multi-period data collection method to minimize common method bias, the method could not truly detect the causality between variables. Future research is suggested to adopt experimental methods or longitudinal data collection methods to further confirm the causal relationship between variables. Moreover, due to the limitations of time and funding, the number of samples collected by the research was limited. Future studies are recommended to collect more samples in other cultural backgrounds to improve the universality of the research results. Thirdly, this study only examine one of the employee’s ability—leadership potential’s effect on leadership ostracism behavior. However, there are other factors which could lead to leadership ostracism behavior. Future studies are suggested to explore the effects of other factors and the interactions. Lastly, this study only explored the negative behaviors of leaders triggered by envy toward employees’ leadership potential. However, leaders may also take some positive actions, such as self-improvement, active learning, and employee coaching. Therefore, future research is suggested to explore the effects of employees’ leadership potential on the positive behaviors of leaders.

## 6. Conclusions

The purpose of this study was to explore the mechanism of employee’s leadership potential on leadership ostracism behavior. The analysis results of the data of 221 employee–leader pairs, collected at three time points, revealed that employees’ leadership potential had a significant positive impact on leadership ostracism behavior and leader’s envy; leader’s envy had a significant positive impact on leadership ostracism behavior and played a full mediating role in the relationship between employees’ leadership potential and leadership ostracism behavior. In addition, employee’s political skills weakened the indirect effect of employees’ leadership potential on leadership ostracism behavior via leader’s envy; the effect of leadership potential on leadership ostracism behavior of employees with more political skills tended to be weaker. This study constructed a theoretical model of the relationships between employee’s leadership potential, leader’s envy, leadership ostracism behavior, and employee’s political skills. These findings expanded the scope of research on employee’s leadership potential, enriched the research on the mediating mechanism of employee’s leadership potential, and uncovered the moderating role of employee’s political skills, which revealed the positive side of political skills and proved prior researches. These findings are inspirational for practical implications as well. Organizations should mitigate the negative outcomes of the leader’s envy and leadership ostracism behavior by carefully appointing managers; recruiting employees with more political skills, implementing corresponding training to improve the political skills, providing training, conducting workshops and socialization, and adjusting the leader’s salary structure. Leaders are suggested to reduce their envy and develop the ability of empathy by personality development trainings. Employees should promote their political skills and improve interpersonal relationships to minimize the harmful effects.

## Figures and Tables

**Figure 1 ijerph-17-03080-f001:**
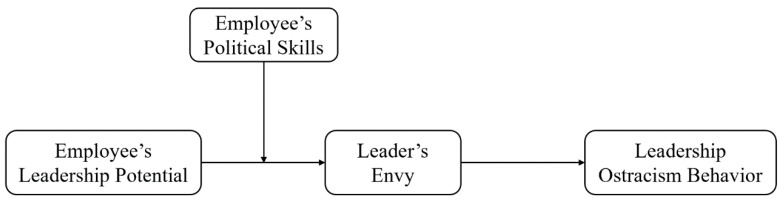
Conceptual model.

**Figure 2 ijerph-17-03080-f002:**
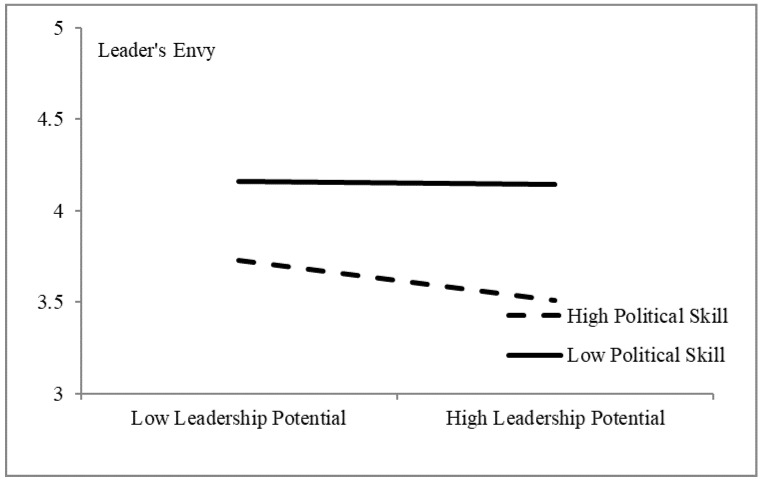
Moderating effects of political skill on leadership potential and leader’s envy.

**Table 1 ijerph-17-03080-t001:** Demographics of respondents.

Participants	Variables	Characteristics	Frequency	Percent (%)
Employees	Gender	Male	128	57.9%
		Female	93	42.1%
	Age	Under 30	120	54.3%
		21–40	77	34.8%
		41–50	17	7.7%
		Over 50	7	3.2%
	Education level	High school certificate or below	22	10.0%
		Technical school and Undergraduate degree	163	73.8%
		Master or higher degree	36	16.3%
Employees	Gender	Male	25	62.5%
		Female	15	37.5%
	Age	Under 30	4	10.0%
		21–40	19	47.5%
		41–50	14	35.0%
		Over 50	3	7.5%
	Education level	High school certificate or below	1	2.5%
		Technical school and Undergraduate degree	32	80.0%
		Master or higher degree	7	17.5%

**Table 2 ijerph-17-03080-t002:** Test results for reliability and validity of the variables.

Variables	Cronbach’s α	CR	AVE
Leadership Potential	0.873	0.874	0.635
Leader’s Envy	0.881	0.882	0.651
Leadership Ostracism Behavior	0.877	0.877	0.546
Employee’s Political Skill	0.975	0.975	0.687

Note. CR = Composite Reliability; AVE = Average Variance Extracted.

**Table 3 ijerph-17-03080-t003:** Results for confirmatory factor analysis.

Model	χ^2^	df	RMSEA	RMR	CFI	TLI
NULL MODEL	6584.229	496	0.236	1.279	0.000	0.000
Four-Factor Model	839.764	458	0.062	0.112	0.937	0.932
Three-Factor Model	1397.113	461	0.096	0.251	0.846	0.835
Two-Factor Model	1509.918	463	0.101	0.254	0.828	0.816
Single-Factor Model	1827.482	464	0.116	0.268	0.776	0.761

Note. **χ^2^** = Chi-square; df = Degrees of Freedom; RMSEA = Root Mean Square Error of Approximation; RMR = Root of the Mean Square Residual; CFI = Comparative Fit Index; TLI = Tucker-Lewis Index.

**Table 4 ijerph-17-03080-t004:** Means, standard deviations, and correlation coefficients of variables.

Variables	Mean	SD	1	2	3	4
Leadership Potential	3.516	1.365	-			
Leader’s Envy	4.305	1.258	0.428 **	-		
Leadership Ostracism Behavior	4.302	1.217	0.288 **	0.380 **	-	
Employee’s Political Skill	3.524	1.383	−0.776 **	−0.522 **	−0.322 **	-

Note. n = 221. ** *p* < 0.01.

**Table 5 ijerph-17-03080-t005:** Results for the effect of leadership potential on leadership ostracism behavior.

	Unstandardized Coefficients		Standardized Coefficients	t	Sig.
	B	SE	Beta		
(Constant)	3.397	0.217		15.623	0.000
Leadership Potential	0.257	0.058	0.288	4.458	0.000
F	19.878 ***				
R^2^	0.083				

Dependent Variable: Leadership ostracism behavior; ***. Correlation is significant at 0.001 level.

**Table 6 ijerph-17-03080-t006:** Results for the effect of leadership potential on leader’s envy ostracism.

	Unstandardized Coefficients		Standardized Coefficients	t	Sig.
	B	SE	Beta		
(Constant)	2.920	0.212		13.749	0.000
Leadership Potential	0.394	0.056	0.428	6.999	0.000
F	48.984 ***				
R^2^	0.183				

Dependent Variable: Leader’s envy; ***. Correlation is significant at 0.001 level.

**Table 7 ijerph-17-03080-t007:** Results for the effect of leader’s envy on leadership ostracism behavior.

	Unstandardized Coefficients		Standardized Coefficients	t	Sig.
	B	SE	Beta		
(Constant)	2.718	0.271		10.030	0.000
Leader’s Envy	0.368	0.060	0.380	6.087	0.000
F	37.051 ***				
R^2^	0.145				

Dependent Variable: Leadership ostracism behavior; ***. Correlation is significant at 0.001 level.

**Table 8 ijerph-17-03080-t008:** The results of the mediating effect for leader’s envy.

Path	Effect	SE	LLCI	ULCI
Total effect				
Leadership Potential→Leadership Ostracism Behavior	0.257	0.058	0.144	0.371
Direct effect				
Leadership Potential→Leadership Ostracism Behavior	0.137	0.061	0.017	0.258
Indirect effect				
Leadership Potential→Leadership Ostracism Behavior	0.120	0.037	0.051	0.195

Note. SE = Standard Error; LLCI = Lower Level Confidence Interval; ULCI = Upper Level Confidence Interval.

**Table 9 ijerph-17-03080-t009:** The results of moderating effect.

Dependent Variable: Leader’s Envy	Coefficient	SE	*t*	*p*	LLCI	ULCI
Constant	4.153	0.094	44.357	0.000	3.968	4.337
Leadership Potential	−0.006	0.087	−0.071	0.943	−0.177	0.164
Political Skill	−0.534	0.091	−5.858	0.000	−0.713	−0.354
Leadership Potential × Political Skill	−0.105	0.041	−2.535	0.012	−0.187	−0.023

Note. SE = Standard Error; *t* = t Value; *p* = Obtained Significance Value; LLCI = Lower Level Confidence Interval; ULCI = Upper Level Confidence Interval.

**Table 10 ijerph-17-03080-t010:** The results of moderated mediator.

Political Skills	Effect	Boot SE	Boot LLCI	Boot ULCI
M-1SD	0.042	0.044	−0.056	0.125
M	−0.002	0.045	−0.102	0.078
M+1SD	−0.046	0.056	−0.172	−0.051

Note. SE = Standard Error; LLCI = Lower Level Confidence Interval; ULCI = Upper Level Confidence Interval; M-1SD = Mean–1 Standard Deviation; M = Mean; M+1SD = Mean+1 Standard Deviation.

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
