# Peer review of "How Employee’s Leadership Potential Leads to Leadership Ostracism Behavior: The Mediating Role of Envy, and the Moderating Role of Political Skills"

_ijerph, 2020, doi:10.3390/ijerph17093080_

Round 1

Reviewer 1 Report

Both employee's leadership potential and leader’s ostracism are emerging areas for researchers and practitioners. I read the manuscript thoroughly and found it rigorous and readable for the readers. However, based on my understanding, I have the following few recommendations which will improve the quality of the manuscript. 

  1. As the authors have mentioned that, “The purposes of this study are as follows: (1) to investigate the impact of employee’s leadership potential on organizational ostracism;”  To make the study more specific, the purpose of the study could be revised as the current study investigates the impacts of employee’s leadership potential on “perceived leader’s ostracism” instead of “organizational ostracism” which is in Line 50 of the manuscript.
  2. Similarly, the authors say that “to analyze the influence of leader’s political abilities on the above relationships from the perspective of individual employees and the boundary conditions of the impact that employee’s leadership potential has on leader ostracism”  kindly recheck the line 53. The current study has incorporated the political skills of employees as a moderating variable in the relationship instead of the leader’s political abilities. Hence, the purpose of the study could be revised in a more clear way.
  3. The managerial implications of the study could be approved by considering other organizational aspects including training, conducting workshops and socialization in order to mitigate the negative outcomes of the manager’s ostracism and manager’s envy. Similarly, the authors can discuss how the organizations incentivize the significance of employee’s leadership potential and employee’s political skills for the betterment of the working environment and interpersonal relationships.

Reviewer 2 Report

Thank you, dear author, for the interesting paper. Here I am giving my major queries of the paper. 

What is the rationality of the study? 

Clearly mention the logic and purpose of the study?

What about the theoretical distribution? There is no theoretical discussion as well as a very limited literature review. 

Broadly elaborate the sample distribution in a table. 

Variable measurement is not clear to me please check it. 

Why you introduced both moderating and mediating impact in the analysis there is no discussion, therefore it is necessary to discuss both issues according to the prior literature (see Masud et al. 2019). 

How do you implement the moderating and mediating effects?

Why political skill has a negative impact while it has both results in the prior study (see references).

Result and discussion are limited in the line of hypothesis also inadequate current citations. 

The conclusion is very short while there are no implications (theoretical and managerial). 

No future direction and limitation mentioned. 

It is required to check the above issues along with the academic research style. 

References: 

Masud, M.A.K; Harun, M; Khan, T; Bae, S; and Kim, J.D. (2019). Organizational Strategy and Corporate Social Responsibility: The Mediating Effect of Triple Bottom Line, https://doi.org/10.3390/ijerph16224559

Hung, M.; Kim, Y.; Li, S. Political connections and voluntary disclosure: Evidence from around the world. J. Int. Bus. Stud. 2018, 49, 272–302.

Reviewer 3 Report

  1. The study explores an interesting and exciting topic. Indeed, the leadership potential often has an effect on “leader ostracism”, but the authors’ model is too simplistic. They do not take into account either the personality of the leader (success-oriented managers are not primarily afraid of their own future, but specifically look for talent for the successful future of their organization) or the age of the leader (if the leader plans a succession in the future, then the leadership potential is not a disadvantage, but an advantage in the leader-employee relationship). The roles of these factors need to be analyzed in detail in terms of their impacts on the relationships.
  2. Literature review is not deep enough in many places. The corroboration of H1 and H2 is weak, there is a need to explain in more detail why employees’ leadership potential affects the leader ostracism (in my opinion, the English expression of the latter variable is not even correct). Examples should also be given of how the leader perceives the employees’ leadership potential and, if so, what the manager can do to hinder this employee’s success. It is also necessary to address what is the situation about the leadership potential of the informal leaders, whose leadership potential has already been accepted by the majority of the other employees in the team. How do they react to the envy of the leader?
  3. H4 is one of the most important results according to the authors, yet its argumentation and logical derivation based on the literature analysis is completely lacking.
  4. In Figure 1, why “Supervisor’s Envy” is the one of the variables instead of the “Leader’s Envy”? This is confusing as the authors use different words compared to their hypothesis, which gives the impression that these two concepts are different.
  5. The statistical methods and interpretation of results are appropriate.
  6. The current Conclusions only summarize the results of the study; this is the function of the abstract. Here you should write about practical advices that can be drawn from the results. What should practicing managers do to ensure that a lower-level leader does not oppress the employees with leadership potential out of mere envy or fear? Is education and training a solution to this problem? Are personality development trainings able to prevent this leadership weakness?
  7. The journal is not primarily concerned with management issues, but (partly) with the impact of workplace problems and challenges on mental and physical health. These aspects are completely missing from the study.

Round 2

Reviewer 2 Report

Thank you, dear author, for the effort to improve the paper. Still, I think you have to discuss the effects of moderating and mediating along with both implications (see Masud et al. 2019). I agree with your arguments but the implication of both effects has some special meanings as well very difficult at a time in an organization. 

Political skill is a debatable matter but this totally depends on the cultural impact and government initiatives. Political skills moderate many ways to the organizational dimension even to control of corruption (see Masud et al. 2019). Therefore,  I think you could critically discuss the issues in the line of your country. 

Other than everything is very well documented. Please critically write these two issues in the line of the references. 

Masud, M.A.K; Harun, M; Khan, T; Bae, S; and Kim, J.D. (2019). Organizational Strategy and Corporate Social Responsibility: The Mediating Effect of Triple Bottom Line, https://doi.org/10.3390/ijerph16224559.

Masud, M.A.K; Bae, S; Javier, M; & Kim, J.D. (2019). Board Directors’ Expertise and Corporate Corruption Disclosure: The Moderating Role of Political Connections, Sustainability,11 (16), 4491.

Reviewer 3 Report

Thank you for your fast and thorough revision!